# Crystal Growth, Thermal and Spectral Properties of Er: LGGG Crystal

**DOI:** 10.3390/ma15144819

**Published:** 2022-07-11

**Authors:** Qiaoyun Chi, Lei Liu, Xianhui Xin, Xiuwei Fu, Zhitai Jia, Xutang Tao

**Affiliations:** State Key Laboratory of Crystal Materials, Shandong University, Jinan 250100, China; 202012830@mail.sdu.edu.cn (Q.C.); 201912614@mail.sdu.edu.cn (L.L.); 202012902@mail.sdu.edu.cn (X.X.); txt@sdu.edu.cn (X.T.)

**Keywords:** mid-infrared laser, Er: LGGG, Czochralski method

## Abstract

A high-quality Er^3+^-doped (Gd_1−x_Lu_x_)_3_Ga_5_O_12_ (Er: LGGG) laser crystal with a size of *Φ* 36 × 45 mm^3^ was successfully grown by the Czochralski (Cz) method for the first time. The effective segregation coefficient of Er^3+^ was determined to be 0.97, close to 1, and, thus, the uniform high-quality Er: LGGG crystal can be grown. In addition, the thermal and spectroscopic properties of Er: LGGG were investigated. Based on the measured characteristics, the Er: LGGG crystal has a huge potential for use in the 3.0 µm mid-infrared laser because of its outstanding optical quality, extraordinary thermal conductivity and stable structure.

## 1. Introduction

In the past decades, mid-infrared (MIR) lasers have been a hot topic in fields such as biomedicine, spectral detection and space military [1,2,3,4]. Currently, one of the main approaches for MIR laser output is the rare-earth lasers, for instance, based on Tm^3+^, Pr^3+^, Dy^3+^, Er^3+^ and Ho^3+^. Among them, Er^3+^-doped crystals show promise in producing commercial 2.7~3.0 µm lasers because they can be efficiently pumped by laser diodes (LDs) with wavelengths of 795 or 975 nm. In addition, the Er^3+^-lasers are likely to own advantages of high power due to the large emission cross section. As is known to all, the 3.0 µm MIR laser can also be applied as a pump sources for obtaining far-infrared tunable lasers from optical parametric oscillator systems [5,6]. Therefore, developing high-quality 3.0 µm MIR Er^3+^-laser crystals is of great significance.

Up to now, there are many reported Er^3+^-doped matrix crystals with the spectral range of around 3.0 µm, such as Y_2_O_3_ [7], Lu_2_O_3_ [8], Y_3_Al_5_O_12_ (YAG) [9], Gd_3_Ga_5_O_12_ (GGG) [10], Lu_3_Ga_5_O_12_ (LuGG) [11], Y_3_Sc_2_Ga_3_O_12_ (YSGG) [12], and CaF_2_ [13]. Among them, the garnet-type Er: GGG crystal is considered as a possible solid-state laser medium because of the advantages of excellent thermal conductivity, excellent physicochemical properties in combination with easy fabrication of large size crystal [14,15]. Er: GGG crystallizes in cubic structure with space group *Ia*3*d*. Both Er^3+^ and Gd^3+^ occupy the dodecahedral coordinated lattice site. As the radius of Gd^3+^ (1.053 Å) is greater than that of Er^3+^ (1.004 Å), the as-grown crystal showed a non-uniform chemical composition distribution due to the segregation effect, which is very harmful to laser operations. One possible approach to overcome this problem is to engineer the site of crystal structure to match the Er^3+^ size. The Lu^3+^ (0.977 Å) has a smaller radius than Er^3+^, so that the modified [Lu_x_Gd_1−x_] dodecahedral site could be suitable for Er-doping. Accordingly, Lu^3+^ have the advantages of optimizing crystal structure and reducing lattice distortion. However, as far as we know, no Er: (Lu_x_Gd_1−x_)_3_Ga_5_O_12_ (Er: LGGG) co-doped crystals have been reported.

The growth of Er: LGGG crystals using the Cz method had not been successfully documented before this work. In the present study, the doping concentrations of Er^3+^ and Lu^3+^ were identified by X-ray fluorescence (XRF). The thermal and optical properties were essential factors in evaluating the feasibility of experiments and were therefore systematically investigated.

## 2. Experimental

### 2.1. Crystal Growth

The Er: LGGG polycrystalline materials with doping levels of 5% Er and 10% Lu were synthesized by the normal solid-state reaction. The source materials in this process were 4N Er_2_O_3_, Lu_2_O_3_, Gd_2_O_3_ and Ga_2_O_3_ powders (Haipuri Rare Earth Materials Company, Changchun, China). These source materials were weighed stoichiometrically. A 2 wt.% excess of Ga_2_O_3_ was added due to decomposition and volatilization when the experiment was carried out. After the powders were mixed evenly, the powders were pressed into several columns and sintered in air at 1350 °C for 40 h.

The Cz method was used to grow the Er: LGGG crystal. The synthesized polycrystalline materials were transferred to an Ir crucible (*Φ* 60 × 60 mm^3^). After all raw materials were melted, a single crystal was pulled with an [111] oriented GGG seed under a rotation rate of 10–15 rpm and a pulling rate of 0.79 mm/h. Argon atmosphere was applied during growth to prevent the oxidation of Ir crucible. When the crystal was taken out, it needed to be annealed at 1500 °C for 15 h in the air.

### 2.2. Characterization Techniques

The successfully grown crystal was processed into crystal samples for characterizations. Single crystal powders were placed in BrukerAXS D2 ADVANCE X-ray diffractometer fitted out Cu-K*α* radiation for measuring X-ray diffraction (XRD). Laue back-reflection was performed on Back-Reflection Laue apparatus (Multiwire MWL 120 with North star software) to further make an assessment of the crystalline quality. A square sample of 4 × 4 × 1 mm^3^ with the thickness in [111] direction was used for measurement. The concentrations of Er^3+^ and Lu^3+^ were analyzed using the XRF technique (Rigaku, ZSX primus II, Tokyo, Japan).

The thermal expansion was characterized using a Mettler–Toledo thermal-mechanical analyzer (TMA/SDTA840). The sample dimensions were 4 × 4 × 4 mm^3^ and the (111) end-faces were fine polished. Specific heat was analyzed using a differential scanning calorimetry (DSC) 8000 differential scanning calorimeter. The grown crystal was processed into 4 × 4 × 1 mm^3^ for the test. A same size sample was gilded and measured in a Netzsch Nanoflash model LFA 457 apparatus for thermal diffusivity. The crystal density was measured by the Archimedes method in distilled water at 25 °C. On the basis of the data of specific heat, thermal diffusivity as well as crystal density, the thermal conductivity data were analyzed.

The absorption spectrum was measured by a Cary 7000 equipment. The absorption cross section was calculated on the basis of the measured absorption spectrum data. The fluorescence spectrum measurements were measured using a Hitachi F4500 spectrophotometer. The fluorescence attenuation curve was measured by single photon counting method, and the fluorescence lifetime was fitted according to the fluorescence attenuation curve. All spectral characteristics were evaluated for the polished (111) plates sized 4 × 4 × 1 mm^3^.

## 3. Results and Discussion

### 3.1. Crystal Growth

The Cz method was used to grow high-quality Er: LGGG crystal with maximum dimensions of *Φ* 36 × 45 mm^3^, as shown in Figure 1. The crystal was free of cracks and inclusions. After polishing, the crystal has high transparency and pale pink color, as shown in Figure 2a. The crystal surface is covered with erosion streaks due to the re-melting during the growth process [16].

### 3.2. Structural Properties

The XRD pattern of Er: LGGG crystal is displayed in Figure 3. As can be seen, it matches well with the GGG pattern [4] (JCPDS 13-493) simulated from crystal structure, indicating that the single phase was obtained. On the basis of XRD data, the lattice parameters were calculated as *a* = *b* = *c* = 12.369 ± 0.002 Å by software FullProf. The value is smaller than of GGG (12.376 Å) [17] because of smaller radius of the Er^3+^ and Lu^3+^ in reference to that of Gd^3+^.

The structural quality of Er: LGGG crystal was characterized by the Laue back-reflection measurements. It is obviously seen from Figure 2b that the diffraction spots are clear and bright, and have good symmetry, which shows a high crystalline quality.

### 3.3. Effective Segregation Coefficient

The effective segregation coefficient of crystal can be calculated on the basis of the formula [18]:(1)keff=CsCl
where *C**_S_* and *C**_l_* are the ion doping concentrations in the crystal and in raw materials, respectively. Based on the XRF data, the effective segregation coefficients of Er^3+^ and Lu^3+^ in Er: LGGG crystals were calculated to be 0.97 and 1.38, respectively. The effective segregation coefficient of Er^3+^ is approximately equal to 1, which is necessary for rapid and high-quality crystal growth. In contrast, the reported value of Er^3+^ in 10 at.% Er: GGG is as high as 1.56, far from idea value of 1 [19]. Therefore, the lattice engineering for Er: LGGG in this work is very successful.

### 3.4. Crystal Density

Three samples were taken from different parts of the crystal and the density was measured by buoyancy method as shown in Table 1 (*m*_0_ is the sample weight in air, *m*_1_ is the sample weight in deionized water, $\rho_{water}$ is the density of the deionized water at room temperature, $\rho_{exp}$ is the density of the sample, and $\rho_{av}$ is the average density of the sample). The average density of Er: LGGG crystal measured is 7.15 ± 0.02 g cm^−3^ at room temperature, larger than 7.088 g cm^−3^ [20] of GGG crystal due to the more heavy atomic masses of Er^3+^ and Lu^3+^. Then, the density value of the crystal at different temperatures can be obtained by the following equation [21]:(2)$$\rho_{T}=\frac{MZ}{V_{T}N_{A}}=\frac{\rho_{T_{0}}}{1+3\alpha \times (T-T_{0})}$$
where $\rho_{T}$ is the crystal density at *T*, *M* is the molar weight of the crystal, *Z* is the number of molecules in the unit cell, *V_T_* is the unit cell volumes at *T*, $\rho_{T_{0}}$ is the density at room temperature of 300 K, and *α* is the thermal expansion coefficients. The average linear expansion coefficient calculated in this work was *α* = 7.7096 × 10^−6^ K^−1^. The density change of Er: LGGG crystal in the temperature range of 298~773 K is shown in Figure 4. With temperature increasing, the density decreases linearly from 7.15 g cm^−3^ at 298 K to 7.07 g cm^−3^ at 773 K.

### 3.5. Thermal Properties

Thermal expansion coefficient is a vital physical property of crystal. According to the Neumann principle [22], the thermal expansion coefficients [α_ij_] are defined by symmetrical second-rank tensor, and *α*_11_ = *α*_22_ = *α*_33_ for Er: LGGG because its structure symmetry is related to cubic system. The temperature-dependent thermal expansion curve of Er: LGGG crystal is presented in Figure 5. The average thermal expansion coefficient was fitted to be 7.71 × 10^−6^ K^−1^, comparable with 9.03 × 10^−6^ K^−1^ of GGG crystal [23].

For laser crystal, the specific heat plays a vital part in affecting the optical damage threshold. Generally speaking, the higher specific heat leads to higher laser damage threshold. The scaling relationship between specific heat and temperature is shown in Figure 6. The specific heat of Er: LGGG crystal increases significantly with temperature from 0.36 J g^−1^ K^−1^ at 293 K to 0.47 J g^−1^ K^−1^ at 573 K, which is slightly lower than that of GGG crystal [20,24], making Er: LGGG crystal perform better at strong pumping.

The thermal diffusivity is a symmetric second-rank tensor. For the cubic system, there is only one independent principal component. Figure 7a shows the thermal diffusivity of Er: LGGG crystal in terms of temperature. It drops from 2.12 mm^2^ s^−1^ at 293 K to 0.81 mm^2^ s^−1^ at 773 K. Then, the formula for deriving the thermal conductivity *κ* can be expressed as follows [25]:(3)$$\kappa =\lambda \cdot \rho \cdot C_{p}$$
where *λ*, *ρ* and *C**_p_* are the thermal diffusivity, density and specific heat of the crystal, respectively, at the corresponding temperature. According to Equation (3), the change of thermal conductivity from 5.43 W m^−1^ k^−1^ at 293 K to 3.28 W m^−1^ k^−1^ at 773 K is shown in Figure 7b, slightly lower than 7.05 W m^−1^ k^−1^ of GGG crystal at room temperature [20].

### 3.6. Spectral Properties

The absorption cross section of Er: LGGG crystals within the limits of 350–1800 nm is shown in Figure 8. Seven strong-absorption bands of Er^3+^ centered at 381, 524, 654, 790, 966, 1471 (1532) nm were observed, and they can be attributed to the transition from the ground state ^4^I_15/2_ to ^4^G_9/2_, ^5^S_3/2_, ^4^F_9/2_, ^4^I_9/2_, ^4^I_11/2_ and ^4^I_13/2_. The absorption cross section *σ* can be expressed as follows [19]:(4)$$\sigma =\frac{\alpha }{N_{C}}=\frac{A}{N_{C}L\times \mathrm{lg}e}$$
where *α* is the absorption coefficient, *N*_c_ is the concentration of doping ions, *A* is the absorbance and *L* is the thickness of the polished crystal. The maximum absorption cross section of Er: LGGG at the peak of 967 nm is 5.28 × 10^−21^ cm^2^, and its full width at half maximum (FWHM) is about 11 nm. The 967 nm band overlaps well with the commercially available high-power InGaAs LDs. The absorption cross-sectional area of Er: YSGG crystal is 1.56 × 10^−21^ cm^2^ [26], 3.4 times lower than that of Er: LGGG crystal. Therefore, the large absorption bandwidth of Er: LGGG crystal is suitable for pumping and can reduce the temperature dependence for the LDs.

Figure 9 is the MIR emission spectrum recorded under the excitation at 967 nm. Two strong fluorescence peaks near 2650 and 2820 nm were observed. The FWHM of MIR emission spectrum are more than 100 nm, indicating the potential to realize tunable and short pulse laser. Furthermore, there is an obvious sub-level jump from ^4^I_11/2_ to ^4^I_13/2_ in Er^3+^. These phenomena fully demonstrated that Er: LGGG crystal shows a promise for the use in the laser field.

Under the excitation of 970 nm pulsed laser, the fluorescence decay curve of Er: LGGG for the upper (^4^I_11/2_) laser level was characterized at 2650 nm. In the case of 10 at.% Er: GGG crystal, the fluorescence decay curve fits a single exponential [19]. However, the doping of Lu^3+^ makes the fluorescence decay curve double exponential, as seen in Figure 10. The fitting values are $\tau_{1}$ = 0.063 and $\tau_{2}$ = 0.805. Then, through the formula the fluorescence lifetime τ can be expressed as follows:(5)$$\tau =\frac{A_{1}\tau_{1}^{2}+A_{2}\tau_{2}^{2}}{A_{1}\tau_{1}+A_{2}\tau_{2}}$$
where *A*_1_ is 0.813, $\tau_{1}$ is 0.063, *A*_2_ is 0.150, $\tau_{2}$ is 0.805. The fluorescence lifetime was evaluated to be 0.584 ms (^4^I_11/2_).The high concentration of Er^3+^ takes a leading part in the delay of fluorescence intensity. They can realize the rapid pumping of the number of particles in the lower level and recycle the energy by the complex up-conversion and cross-relaxation processes between adjacent ions. The growth of Er-doped crystals with high concentration needs further development.

## 4. Conclusions

In summary, the bulk Er: LGGG crystal with superior quality was successfully grown using the Cz method. The dimension was *Φ* 36 × 45 mm^3^. According to the XRF measurements, the effective segregation coefficient of Er^3+^ was 0.97, close to the 1. The thermal and spectral properties were evaluated and analyzed. The coefficient of thermal expansion increases linearly with temperature, and the average linear thermal expansion coefficient was 7.71 × 10^−6^ K^−1^. The specific heat also increased from 0.36 J g^−1^ K^−1^ at 293 K to 0.47 J g^−1^ K^−1^ at 573 K. The thermal diffusivity decreased from 2.12 mm^2^ s^−1^ at 293 K to 0.81 mm^2^ s^−1^ at 773 K. The thermal conductivity was deduced from 5.43 W m^−1^ k^−1^ at 293 K to 3.28 W m^−1^ k^−1^ at 573 K. In addition, absorption spectrum, fluorescence spectrum and fluorescence decay curve of Er: LGGG were measured at room temperature. The fluorescence lifetime was as high as 0.584 ms (^4^I_11/2_) by fitting with the use of single exponential function. The above experimental results proof that the Er: LGGG crystal is a MIR laser material with great potential for applications.

## Figures and Tables

**Figure 1 materials-15-04819-f001:**
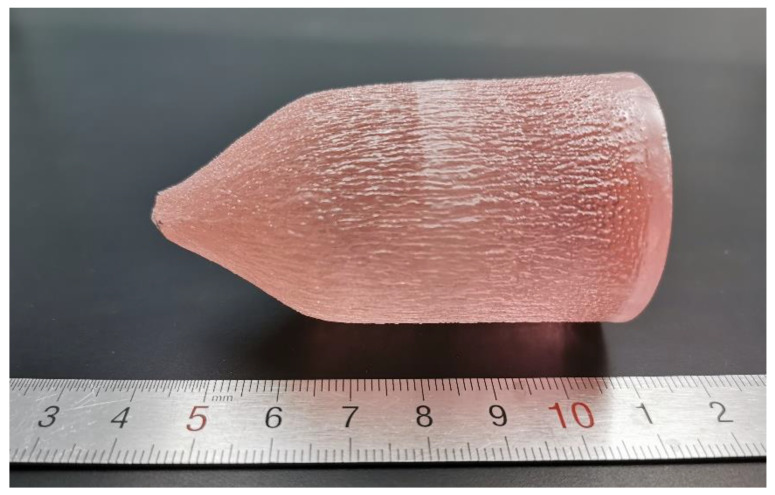
Photograph of as-grown Er: LGGG crystal.

**Figure 2 materials-15-04819-f002:**
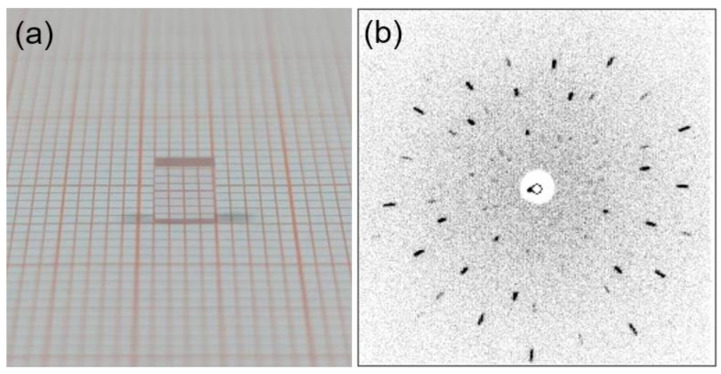
(**a**) The sample (4 × 4 × 4 mm^3^) with a fine (111) end-face polishing used in Laue back-reflection measurements; (**b**) characteristic Laue back-reflection patterns.

**Figure 3 materials-15-04819-f003:**
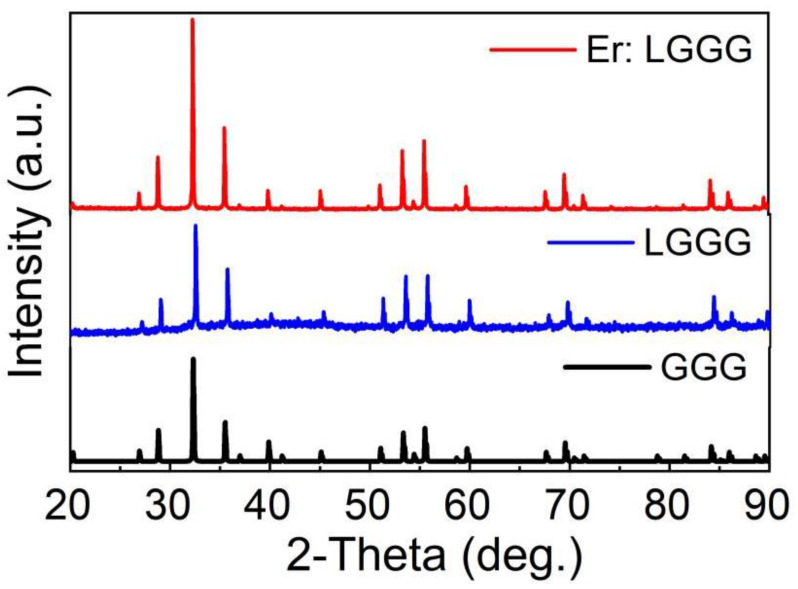
Powder XRD patterns of Er: LGGG and simulated GGG crystals.

**Figure 4 materials-15-04819-f004:**
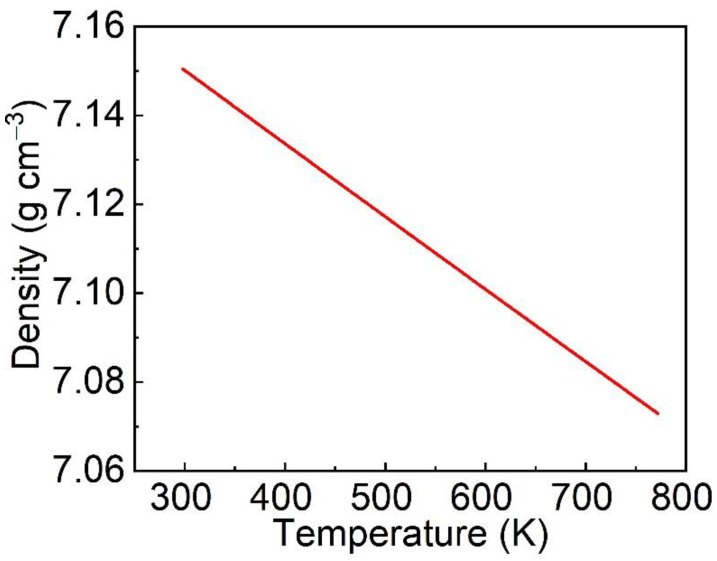
Density of Er: LGGG crystal versus temperature.

**Figure 5 materials-15-04819-f005:**
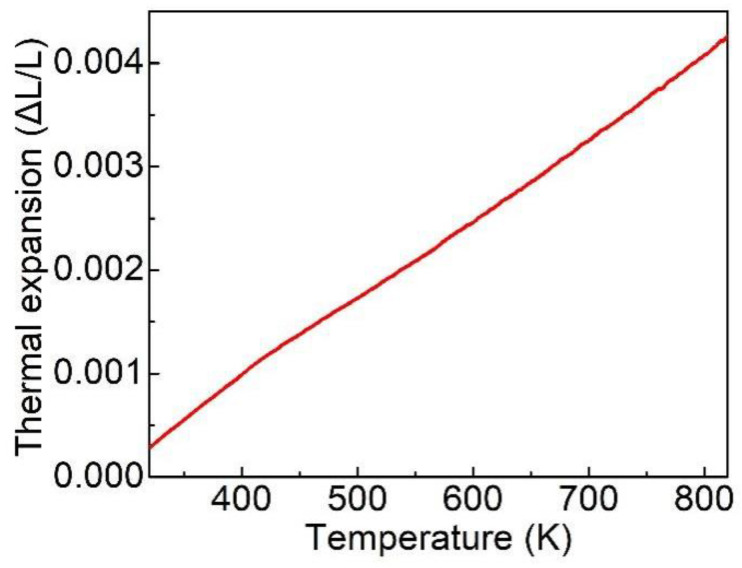
Thermal expansion ratio of Er: LGGG versus temperature.

**Figure 6 materials-15-04819-f006:**
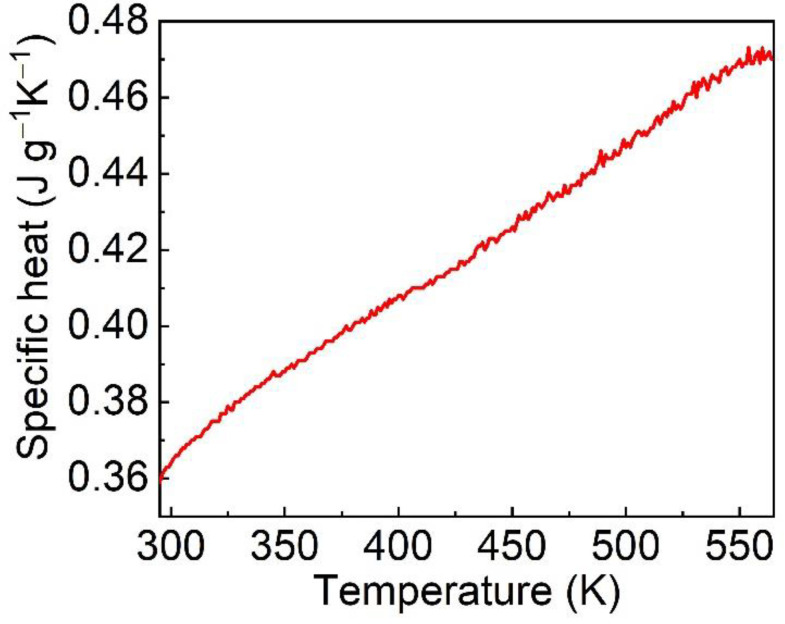
Specific heat of Er: LGGG crystal versus temperature.

**Figure 7 materials-15-04819-f007:**
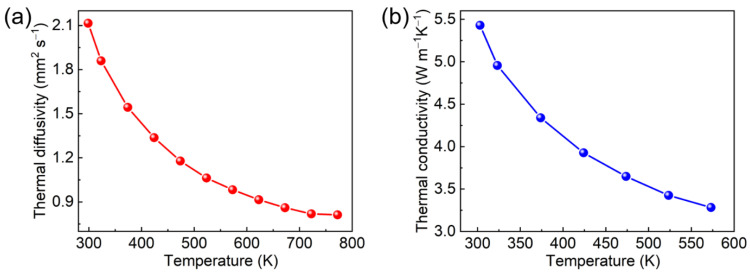
(**a**) Thermal diffusivity and (**b**) calculated thermal conductivity of Er: LGGG crystal in terms of temperature.

**Figure 8 materials-15-04819-f008:**
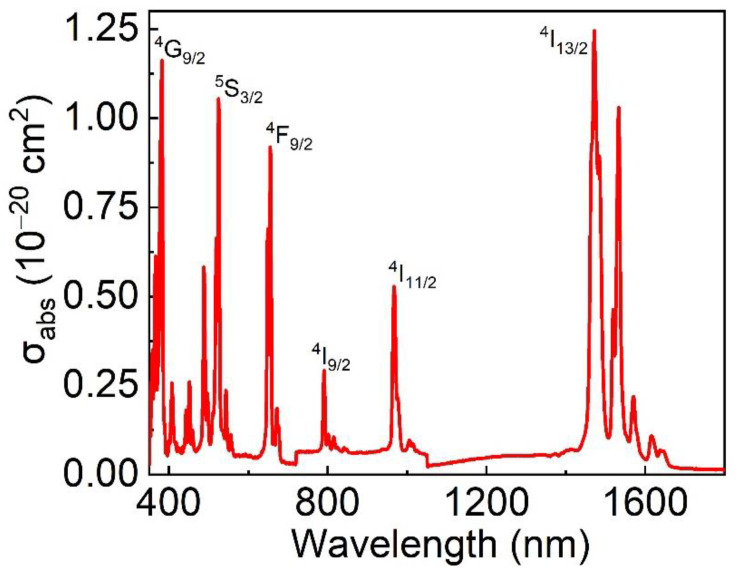
Absorption spectra of the Er: LGGG at room temperature.

**Figure 9 materials-15-04819-f009:**
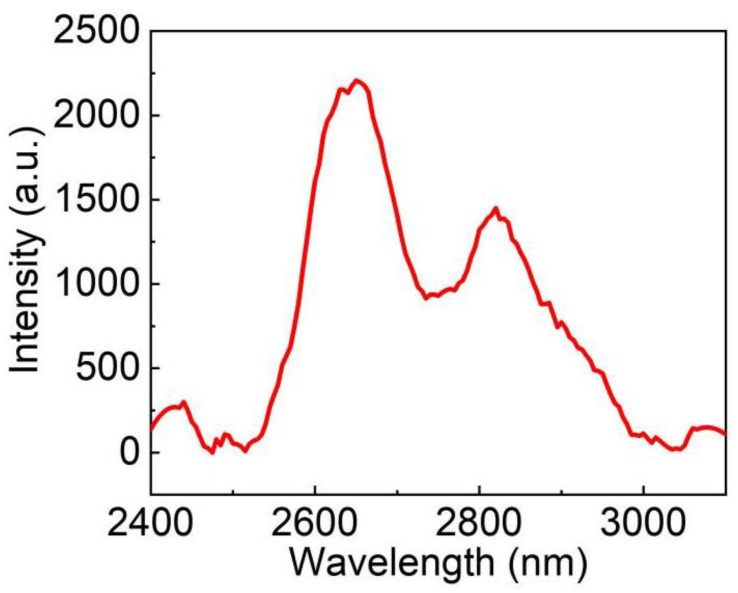
The room-temperature fluorescence spectrum of Er: LGGG under the 967 nm diode excitation.

**Figure 10 materials-15-04819-f010:**
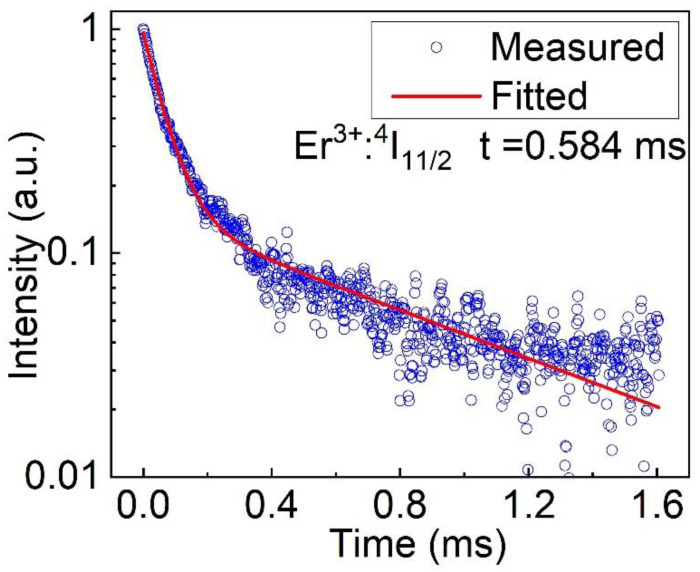
The fluorescence decay curve of Er: LGGG versus time at 2650 nm.

**Table 1 materials-15-04819-t001:** Density of Er-LGGG crystal.

	Sample 1	Sample 2	Sample 3
m_0_ (g)	1.0935	2.4696	3.8052
m_0_ − m_1_ (g)	0.1528	0.3435	0.5324
*ρ*_water_ (g/cm^3^)		0.998	
*ρ*_exp_ (g/cm^3^)	7.1437	7.1768	7.1331
*ρ*_av_ (g/cm^3^)		7.15 ± 0.02	

## Data Availability

The data presented in this study are available on request from the corresponding author.

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
