# Peer review of "Crystal Growth, Thermal and Spectral Properties of Er: LGGG Crystal"

_materials, 2022, doi:10.3390/ma15144819_

Round 1
Reviewer 1 Report
Dear Editor and Authors:
I have carefully read and revised the manuscript by Chi and colleagues entitled “Crystal growth, thermal and spectral properties of Er:LGGG laser crystal”. The authors described the growth and characterization (structural, thermal and spectral) of a novel matrix for Erbium in the mid-IR looking to solve the problems associated to the more conventional laser crystal Er:GGG. The authors provide sufficient details to replicate their work. But for a few aspects (see comments below), the experiments seem to be well conducted, and the analysis is extensive and thorough. The most important aspect I missed was the actual demonstration of laser emission itself. Actually the title is a bit misleading, as one may infer that, indeed, Er:LGGG IS a laser material. Nevertheless, Er:LGGG hasn’t been shown to lase, neither in this manuscript nor in others. It is true that this material has a potential to become a laser crystal, but I wouldn’t abuse of the term laser crystal (as in the title). At the end of the day, a crystal is not a laser material until a laser experiment says so. There are many known facts that can preclude the laser emission even when the weak excitation properties are ideal. Notoriously, the excited state absorption at both the pump and emission regions. In any case, that was not the goal of this manuscript and, if the authors respond to all my queries, I wouldn’t have inconvenience in recommending its publication in Materials MDPI.
Comments:
As a general comment, this manuscript requires a deep English editing. It is mostly understandable, but the style and grammar can be greatly improved.
Fig 2 and lines 95-99: The authors compare the XRD pattern of Er:LGGG with that of the simulated GGG one. I wonder why the authors didn’t simulate LGGG instead.
Lines 147-152: Is the specific heat of GGG at 573 K as well larger than that of LGGG? If not, LGGG could be better at weak pumping but worse at strong pumping. Please, clarify.
Fig 8 shows a strong background, probably due to Fresnel reflection. Actually, it is so strong that it is nearly ¼ of the total contribution to the absorbance of the transition at 967 nm. Then, the reported value of absorption cross section is incorrect. I would suggest subtracting this background to report the correct cross section value. In addition, I would recommend the authors to display in Fig 8 the absorption cross section instead of the absorbance (not absorbancy, by the way).
Lines 180-183: how does the absorption cross section compares to that of GGG? All the paper has used GGG as reference and now they use YSGG. Why?
Figure 9: Please, do a better background subtract or repeat the experiment to have a better signal to noise ratio. It is not admissible to have negative intensities in a fluorescence spectrum. In addition, I would recommend the authors not to use points for the spectrum, but a line.
Line 187: I guess that the FWHM is more than 100 nm, not 10 nm.
Figure 10: I would use a log scale in the y-axis so that the single exponential can be better recognized.
Line 198: What do the authors mean by “delayed fluorescence”? That the lifetime is lengthened? Please, clarify.
In addition, I would like to know which is the luminescence quantum yield, as this is a fundamental parameter for a laser material. On the one hand, it would allow the authors to have an estimate on the stimulated emission cross section and thus delineate whether it would make a good laser material as compared to others, like GGG. On the other hand, every absorbed photon that doesn’t end up emitted is wasted as excess heat, and that will ultimately affect the laser performance, irrespective of the thermal properties of the matrix. The larger the quantum yield, the larger the amplification and the lesser the excess of heat.
Reviewer 2 Report
My comments are in attached file.

Round 2
Reviewer 1 Report
Dear Editor and Authors:
I have carefully read and assessed the revision of the manuscript by Chi and colleagues entitled “Crystal growth, thermal and spectral properties of Er:LGGG”. While most queries and concerns have been solved, one issue remains unresolved, and a new concern has risen (see comments below). Accordingly, I cannot yet recommend publication.
Comments:
Point 9 in authors rebuttal document: The authors followed my advice and represent the decay curve using a log-scale to better distinguish the exponential decay. The problem is that this way of representing the data has revealed that the decay is not a mono-exponential, but, at least, a bi-exponential. I am confident that the authors have fitted an exponential function to the data, but they have surely left the background contribution free. This is, I reckon the authors have used an expression of the kind I(t) = I_0 + I_1*exp(-t/tau). But that it is not the way the fluorescence decays. They should fix I_0 = 0. But if they do that, the curve will not fit the data. Instead, they would have to fit an expression of the kind I(t) = I_1*exp(-t/tau_1) + I_2*exp(-t/tau_2). The fit would be much better, but the author would face a new problem. The decay rate would now have two components (tau_1 and tau_2) with different weights (I_1 and I_2), and then they would have to identify the reason why, at least tentatively. Accordingly, I am afraid that I have to ask the authors to carry out this new analysis and discussion.
Point 11 in authors rebuttal document: In this case the authors have not followed my advice but have provided a non-satisfactory rebuttal. If this paper is reporting the photophysical properties of a new material, then they have to provide information on the quantum yield in this paper, not in a future one. The statement “Producing light and analyzing its cause will be the focus of our next study.“ could be applied to the laser emission but not to fluorescence, which is reported in this work. Again, I am afraid that I must strongly ask the authors to measure or estimate the fluorescence quantum yield.
